# Why do Vision Transformers Have Better Adversarial Robustness than CNNs?

## Abstract

Deep-learning models have performed excellently in various fields because of advances in computing power and the large-scale datasets used to train large models. However, they have an inherent risk that even a small change in the input can result in a significantly different output of the trained model. Therefore, it is crucial to evaluate the robustness of deep-learning models before we trust the models' decisions. In this paper, we evaluate the adversarial robustness of convolutional neural networks (CNNs), vision transformers (ViTs), and CNNs + ViTs, which are typical structures commonly used in computer vision, based on four new model-sensitivity metrics that we propose. These metrics were evaluated for random noise and gradient-based adversarial perturbations. For a fair comparison, models with similar capacities were used in each model group, and the experiment was conducted separately using ImageNet-1K and ImageNet-21K as the pretraining data. The experimental results showed that ViTs were more robust than CNNs for gradient-based adversarial attacks, and our quantitative and qualitative analysis of these results brings to light the cause of the difference.

## 1 Introduction

Convolutional neural networks (CNNs) have been actively used in a variety of vision tasks, such as classification (He et al., 2016; Krizhevsky et al., 2012; Simonyan & Zisserman, 2015), object detection (Girshick et al., 2014; Liu et al., 2016; Redmon et al., 2016), segmentation (Chen et al., 2018; Ronneberger et al., 2015), and image generation (Choi et al., 2018; Radford et al., 2016; Zhu et al., 2017). In recent years, many studies have adopted transformers (Devlin et al., 2019; Vaswani et al., 2017) in the vision domain; this architecture has previously proven highly effective in natural language processing (NLP). Transformer-based models, represented by vision transformer (ViT), show that using large-scale datasets such as ImageNet-21K (Deng et al., 2009) or JFT-300M (Hinton et al., 2015) rather than ImageNet-1K, which is commonly used for CNN pretraining, results in significant performance improvement. These results indicate that using large-scale datasets is effective in the vision domain as well as in NLP (Dosovitskiy et al., 2021; Steiner et al., 2022). Because of these characteristics, ViTs perform better than CNNs in image classification, object detection, and segmentation task (Steiner et al., 2022; Liu et al., 2021; Ryoo et al., 2021; Touvron et al., 2021).

As the scope of using deep neural networks has increased, researchers have attempted to identify the inherent characteristics of deep learning models. For example, CNNs focus more on texture than shape because of their convolution operation (Geirhos et al., 2019; Hermann et al., 2020; Mummadi et al., 2021; Naseer et al., 2021). In contrast, ViTs are known to focus more on shape than texture because they can learn global interactions through the self-attention mechanism (Naseer et al., 2021). Therefore, ViTs can recognize objects as human beings do rather than like CNNs (Naseer et al., 2021). Various studies have evaluated the robustness by identifying the characteristics of deep-learning models applied in various fields (Naseer et al., 2021; Augustin et al., 2020; Bhojanapalli et al., 2021; Hendrycks et al., 2021; Su et al., 2018). For example, Bhojanapalli et al. (2021) compared the out-of-context performance of CNNs and ViTs on the ImageNet-R dataset, which contains synthesized natural noise from real-world data, and the ImageNet-C dataset, which contains arbitrary corruptions. When pretraining with ImageNet-1K, ViTs performed worse than CNNs, but when pretraining with large amounts of data, such as ImageNet-21K or JFT-300M, the performance of ViTs

and CNNs performed similarly. Evaluating the robustness of the deep learning models requires verifying not only against natural noise, as in Bhojanapalli et al. (2021), but also against adversarial perturbations (Szegedy et al., 2014), which are generated by an adversary. Adversarial examples have a significant impact on the reliability of deep-learning models because they result in misclassification through perturbations that are difficult for humans to recognize (Szegedy et al., 2014). Considering this importance, many studies have been conducted on the adversarial robustness of CNNs (Szegedy et al., 2014; Goodfellow et al., 2015; Kurakin et al., 2017; Madry et al., 2018). In contrast, although some studies have been conducted recently on evaluating the adversarial robustness of ViTs (Bhojanapalli et al., 2021; Benz et al., 2021; Mahmood et al., 2021; Shao et al., 2022), few works have systematically compared and analyzed why the adversarial robustness of CNNs differs from that of ViTs.

In this paper, we aim to determine whether there are any differences in adversarial robustness between CNNs, ViTs, and CNNs + ViTs(hybrids) and to comprehend why these differences occur. We first evaluate the adversarial robustness of the three types of models through adversarial perturbations generated by gradient-based adversarial attacks such as the fast gradient sign method (FGSM; Goodfellow et al. (2015)) and projected gradient descent (PGD; Madry et al. (2018)). To compare the difference in adversarial robustness among the models, we use the infidelity to measure the completeness of an explanation (Yeh et al., 2019). Yeh et al. (2019) assumed that a model with a lower Hessian upper bound might lead to a better infidelity score, meaning that a model with a low infidelity score could have small and smooth gradients with respect to the perturbed input. In addition, we propose four novel metrics based on the max-sensitivity proposed by Yeh et al. (2019) to comprehend this assumption. Our proposed metrics evaluate the sensitivity using inputs comprising (1) the model prediction class, (2) the cross entropy (CE) loss, (3) the derivative of the CE loss, and (4) the model logits. We compare the sensitivities of the models using random noise and adversarial perturbations. We experiment with three perspectives to show the differences in model sensitivity; the model capacity, perturbation radius, and the results using pretrained datasets. Our quantitative and qualitative analyses intuitively show which types of models are more sensitive and what causes the difference.

## 2 Related Work

### 2.1 Vision Transformers

ViT (Dosovitskiy et al., 2021) was proposed to apply the transformer, typically used in NLP (Devlin et al., 2019; Vaswani et al., 2017), to the image processing domain. In contrast to a CNN, which receives an entire image as its input and learns the local information of the image through a convolution operation, a ViT receives an input image divided into patches. Because these patches pass through a multi-head attention (MHA) layer, the ViT can learn global information better than a CNN. As the input patches go through the transformer block of an MHA layer, the distance of the attention score between the heads increases, and the MHA layer learns different representations. Here, the attention distance has a similar role to that of the receptive field in a CNN (Dosovitskiy et al., 2021). Furthermore, in contrast to a CNN, which performs the classification through representations of the input image, a ViT classifies an image through additional tokens called class tokens in addition to the input patches.

### 2.2 Adversarial Robustness of Image Classification Models

#### 2.2.1 Adversarial Attack

An adversarial example is generated by intentionally combining an original input with a perturbation created by an adversary to cause the model to make a mistake (Szegedy et al., 2014). Adversarial perturbations are very difficult for humans to recognize because their magnitude is bounded by a very small value (Szegedy et al., 2014). Because of this characteristic, understanding the robustness of deep-learning models to adversarial attacks is essential for their reliable use in computer vision. The FGSM, a typical adversarial attack, is created by Eq. (1) below:

$$x_{adv} = x + \epsilon sign(\nabla_x L(\theta, x, y)), \tag{1}$$

where $\epsilon$ denotes the perturbation budget, $L$ is the loss function, and $\theta$ represents the model parameters. FGSM is an efficient attack method because it calculates the perturbation in a single step (Goodfellow et al., 2015). PGD, which generates an adversarial example through Eq. (2), creates adversarial perturbation according to the same concept as the FGSM, but it is known to be a more powerful attack:

$$x^{t+1} = \prod(x^t + \alpha sign(\nabla_x L(\theta, x, y))), \tag{2}$$

where $t$ denotes the iteration step, $\alpha$ is the perturbation budget at each step, and $\prod$ denotes the max-norm ball. PGD is calculated in multi-steps when generating a perturbation and has a higher attack success rate than FGSM because it begins the calculation at a random starting point inside the max-norm ball (Madry et al., 2018). In this study, we used the FGSM and PGD to evaluate the adversarial robustness of the image classification models.

### 2.2.2 Adversarial Robustness of Vision Transformers

Recently, studies have been conducted to identify and understand the inherent risks of ViTs by evaluating their adversarial robustness (Bhojanapalli et al., 2021; Benz et al., 2021; Mahmood et al., 2021; Shao et al., 2022). Shao et al. (2022) showed that ViTs are more robust to adversarial attacks than CNNs. Furthermore, Benz et al. (2021), Mahmood et al. (2021), and Shao et al. (2022) evaluated the transferability of adversarial examples between models, showing that ViTs were more robust than CNNs. However, these studies compared the adversarial robustness of models pretrained with different datasets. This comparison is unfair because the performance difference based on the size of the pretraining dataset is more enormous in ViTs than in CNNs (Dosovitskiy et al., 2021). Hence, we evaluated the image classification models pretrained with the same dataset and independently compared the pretraining dataset-based results.

Bhojanapalli et al. (2021) compared the robustness of ViTs and CNNs based on their capacities and pretraining datasets. Moreover, because of the architectural characteristics of ViTs, they analyzed the robustness by comparing the encoding blocks, analyzing the self-attention, and evaluating the correlation of representations between layers. However, they did not provide sufficient demonstrations regarding the disparities between the two models with respect to the magnitudes of adversarial perturbations. In this paper, we compare the results of each model to understand the models' characteristics through quantitative and qualitative analyses with respect to the sensitivity of the various perspectives.

## 3 Robustness Evaluation of Adversarial Attack

### 3.1 Image Classification Models

In this section, we evaluate the adversarial robustness of CNNs, ViTs, and hybrids. Table 1 shows the number of parameters for each model used in our experiments and the datasets used to pretrain the models. For the ViTs, we use the architecture first proposed by Dosovitskiy et al. (2021) and follow their method of denoting the models. For example, "ViT-L/16" means that input patches of size $16 \times 16$ were used in ViTs of a "large" size. We use four model sizes: tiny (Ti), small (S), base (B), and large (L), in order of size. Dosovitskiy et al. (2021) also proposed a hybrid-type model architecture to train a ViT with extracted features using ResNetV2 as the backbone. To obtain various insights, we also evaluate the adversarial robustness of the hybrid model architecture. The hybrids are denoted differently depending on the number of layers in ResNet. For example, when ResNetV2-50 was used as the backbone in ViT-B/16, the model was denoted as "ViT-R50+B/16."

According to Dosovitskiy et al. (2021), ViT performance heavily depends on regularization. Steiner et al. (2022) experimentally compared various regularization methods and proposed a training method that performed best. In this paper, we use ViTs and hybrids pretrained with ImageNet-1K and ImageNet-21K, as used in Steiner et al. (2022), for a fair comparison. The CNNs use big transfer (BiT), proposed by Kolesnikov et al. (2020). BiT has an architecture modified to use ResNetV2 robustly in more diverse situations. In this paper, we use the same pretraining dataset for a fair comparison with the ViTs and configure the models to have as similar a number of parameters as possible. The BiT is also represented according to the model

Table 1: Model architectures used in our experiments.

| Method | #Param. | Datasets |
|---|---|---|
| CNN | | |
| ResNetV2-50x1 | 24M | ImageNet-1K, ImageNet-21K |
| ResNetV2-101x1 | 43M | ImageNet-1K, ImageNet-21K |
| ResNetV2-50x3 | 212M | ImageNet-1K, ImageNet-21K |
| ResNetV2-101x3 | 382M | ImageNet-1K, ImageNet-21K |
| ViT | | |
| ViT-Ti/16 | 6M | ImageNet-1K, ImageNet-21K |
| ViT-S/16 | 22M | ImageNet-1K, ImageNet-21K |
| ViT-S/32 | 23M | ImageNet-1K, ImageNet-21K |
| ViT-B/16 | 86M | ImageNet-1K, ImageNet-21K |
| ViT-B/32 | 88M | ImageNet-1K, ImageNet-21K |
| ViT-L/16 | 303M | ImageNet-1K, ImageNet-21K |
| ViT-L/32 | 306M | ImageNet-21K |
| Hybrid | | |
| ViT-R+Ti/16 | 6M | ImageNet-1K, ImageNet-21K |
| ViT-R26+S/32 | 36M | ImageNet-1K, ImageNet-21K |
| ViT-R50+B/16 | 98M | ImageNet-21K |
| ViT-R50+L/32 | 328M | ImageNet-1K, ImageNet-21K |

architecture. For example, "ResNetV2-50x3" refers to a case where the number of hidden layers is 50, and the ratio of widening the channel of the hidden layer is three.

## 3.2 Robustness Evaluation Metrics

We evaluate the adversarial robustness of CNNs, ViTs, and hybrids through FGSM and PGD, which are attack methods using first-order adversaries. We use the infidelity proposed by Yeh et al. (2019) to evaluate the adversarial robustness quantitatively. Additionally, based on the max-sensitivity (Yeh et al., 2019), we propose DiffPred-SENS, Loss-SENS, LossGrad-SENS, and Logit-SENS. We also analyze the adversarial robustness quantitatively by measuring and comparing how sensitively the models react to random noises and adversarial perturbations.

### 3.2.1 Infidelity

Infidelity shows how well the attribution—calculated through the explanation function for the perturbation $I$, defined as the difference between a particular baseline $x_0$ and the input—matches the output change of the model for perturbation. Infidelity indicates the accuracy of the explanation function by reflecting its completeness, an axiom of the explanation function (Yeh et al., 2019; Sundararajan et al., 2017). For example, the smaller the infidelity, the better the attribution that the explanation function generated represents an important region that influences the model's output. In addition, The Hessian $||\nabla_x^2 f(x + \delta)||$ have proven to be the upper bound of infidelity in Yeh et al. (2019). Therefore, it can be assumed that a model with a low infidelity score has small and smooth gradients for perturbations added in input. In our experiments, we use the vanilla gradient (Simonyan et al., 2014) as an explanation function for the infidelity. The vanilla gradient $\Phi$ refers to the derivative of the input image for $f_c(x)$, the model's confidence. Here, the calculated value for each pixel indicates the size and direction of the attribution for the target class:

$$\Phi = \frac{\partial f_c(x)}{\partial x}, \tag{3}$$

$$\text{INFD}(\Phi, f, x) = \mathbb{E}_{I \sim \mu_I}[(I^T \Phi(f, x) - (f(x) - f(x - I)))^2], \tag{4}$$

### 3.2.2 Proposed Metrics

It is necessary to determine whether the model has small and smoothed gradients to perturbation to evaluate the robustness of the model. However, there are two issues in the process of verifying whether the model has a small Hessian. Firstly, we evaluated which type of model exhibits a small infidelity value. However, just because a model has a small infidelity, it cannot be guaranteed that the corresponding Hessian value has also decreased. There are some methods to directly constrain the Hessian to be small, but they are computationally expensive. As an alternative, Yeh et al. (2019) proposed adversarial training. The expected value of the objective function $\mathcal{L}(f(x + \delta), y)$ in adversarial training was shown to hold Hessian as an upper bound. Hence, adversarial training improved infidelity and showed to be robust to perturbations. However, it cannot be guaranteed that the Hessian upper bound is smaller even if infidelity improves through adversarial training. Therefore, it is challenging to evaluate the robustness of the model solely based on infidelity. Second, directly calculating the expected value of all Hessians to confirm the presence of a small Hessian is computationally expensive. Therefore, in addition to infidelity, we propose four metrics to evaluate the sensitivity to perturbation by checking the presence of small and smooth gradients.

**Loss-SENS**. If two different classifiers misclassify the same noisy image, the robustness of the model can be judged to be different when the loss is different. We propose Loss-SENS to determine this quantitatively. If the Loss-SENS value is large, the model is vulnerable to perturbation because the size of the loss increases even with a small amount of noise:

$$\text{Loss-SENS}(\mathcal{L}, f, \gamma, x, y) = \max_{||\delta|| \leq \gamma} |\mathcal{L}(f, x + \delta, y) - \mathcal{L}(f, x, y)|. \tag{5}$$

**LossGrad-SENS**. LossGrad-SENS measures the gradient change of the model's loss to the input. LossGrad-SENS is the value of the $l_2$ norm of the difference between the gradient of the loss concerning normal input and that of the loss for noisy input. Thus, a model with a small LossGrad-SENS value is robust to input noise. This metric indicates the change ratio to compare different scale for each model:

$$\text{LossGrad-SENS}(\nabla_x \mathcal{L}, f, \gamma, x, y) = \max_{||\delta|| \leq \gamma} \frac{||\nabla_x \mathcal{L}(f, x + \delta, y) - \nabla_x \mathcal{L}(f, x, y)||}{||\nabla_x \mathcal{L}(f, x, y)||}. \tag{6}$$

**Logit-SENS**. The proposed Logit-SENS determines the change in the model's logit vector for inputs with a small amount of noise. A large Logit-SENS means that the model is sensitive to even a small amount of noise. Conversely, if the value is small, it is a robust model with a small difference in the result, even if noise is added to the input:

$$\text{Logit-SENS}(f, x, \gamma) = \max_{||\delta|| \leq \gamma} \frac{||f(x + \delta) - f(x)||}{||f(x)||}, \tag{7}$$

where $f$, $\delta$, and $\gamma$ denote the target model, the noise added to the input, and the radius, respectively. For the same reasons as LossGrad-SENS, Logit-SENS also represents the rate of change.

**DiffPred-SENS**. DiffPred-SENS, which measures the change in model prediction when a small amount of noise is added to the input, counts the cases where the model prediction for the noisy input differs from that based on the original image. A large DiffPred-SENS value means that the model makes a classification prediction different from the prediction based on the original input, even with a small amount of noise, and thus is vulnerable to noise:

$$\text{DiffPred-SENS}(f, \gamma, x) = \frac{1}{N_{noise}} \sum_{i=1}^{N_{noise}} \mathbb{1}(\arg\max f(x + \delta_i) \neq \arg\max f(x)), \tag{8}$$

where $\mathbb{1}$ represents the indicator function.

The four metrics consider the change of perturbation from the perspective of loss and target. Firstly, Loss-SENS and LossGrad-SENS are indices proposed to evaluate the change of loss. Loss-SENS is intended to see how much the model's loss for the input varies with perturbation. LossGrad-SENS indicates the change of gradients of the loss with respect to the input. It is assumed that if the change of loss is small, it is

difficult to optimize the adversarial example to degrade the performance of the model. The same situation is assumed when the change of loss gradients is small. However, it is considered that even if the loss is small, there may be situations where the gradients of loss with respect to the input are relatively large, so both metrics are compared together to see if they show consistent results. Next, Logit-SENS and DiffPred-SENS are metrics for considering the change of target. Even if loss and gradients of loss change greatly, adversarial examples are generated by considering the direction, not the magnitude of gradients, in methods such as PGD and FGSM. That is, even if the magnitude of the change is large, the change of prediction may not occur. Therefore, Logit-SENS and DiffPred-SENS are proposed to consider the change of target. Logit-SENS measures the change in logits. The magnitude of the logit changes when the confidence of the target changes greatly. Therefore, the degree of change in the model's confidence in the target can be determined. However, just because the target's confidence decreases, it does not necessarily mean that the prediction changes. To consider this, DiffPred-SENS evaluates how much the actual prediction has changed through the perturbation by considering the change in the prediction through DiffPred-SENS. Thus, we have quantitatively evaluated the model robustness with response to perturbations based on a total of five metrics.

## 4 Experiments

### 4.1 Experimental Settings

In our experiments, five epsilon values were used to generate random noise and adversarial perturbations: 0.001, 0.003, 0.01, 0.03, and 0.1. In the PGD case, the adversarial examples were generated by setting the number of iterations $t$ to 40 times and $\alpha$ to 2/255. The attacks were conducted under a white-box attack model in which the attacker knows all the information about the target model (Carlini et al., 2019) and a black-box attack in which the model parameters are inaccessible (Papernot et al., 2017). In other words, in the white-box attack scenario, the adversarial robustness of the model was evaluated through the adversarial example generated by calculating Eq. (1) and Eq. (2) through the model parameters. In the black-box attack scenario, unlike the white-box attack scenario, because the attacker does not have access to information about the target model, the robustness of the target model was evaluated with the adversarial example generated through a substitute model, in which the parameters were accessible. This is possible because the adversarial example can be transferred between the models (Szegedy et al., 2014; Goodfellow et al., 2015; Papernot et al., 2016). In our evaluations for the black-box attack, the transfer attack was performed, assuming that the attacker could access the information of the remaining models after excluding the target model. We used the PyTorch adversarial attack library torchattacks (Kim, 2020).

We fine-tuned the models on CIFAR-10 and CIFAR-100 via models pretrained on ImageNet-1K and ImageNet-21K, as shown in Table 1. All the models in Table 2 were fine-tuned in the same environments. Fine-tuning was done with a stochastic gradient descent optimizer using a momentum of 0.9 and a learning rate of 0.003. We also used a cosine annealing scheduler. The models were fine-tuned for 10,000 training steps with a batch size of 128. We resized the input of CIFAR-10 and CIFAR-100 to $224 \times 224$ via bilinear interpolation to match the image size on which the model was pretrained. The pretrained models were loaded from timm, a PyTorch image model library (Wightman, 2019).

In this study, we evaluated the adversarial robustness of the pretrained models using CIFAR-10, CIFAR-100, and the ImageNet-1K datasets. The datasets used for the evaluation were constructed by sampling the same number of images for each class from the test set of each dataset. Specifically, we extracted 1,000, 5,000, and 5,000 images from CIFAR-10, CIFAR-100, and ImageNet-1K, respectively, for evaluation. For example, in the case of CIFAR-10, which has ten target classes, we extracted 100 images for each class. In the experiment, the extracted evaluation images were used to compare the defensive performances against the adversarial attacks compared to the top-1 accuracy of the clean images.

Table 2: Benchmark performance. Columns i1k and i21k (ImageNet-1K and ImageNet-21K) report the top-1 accuracy (%) after fine-tuning.

| Method | ImageNet-1K | | CIFAR-10 | | CIFAR-100 | |
| | i1k Acc(%) | i1k & i21k Acc(%) | i1k Acc(%) | i21k Acc(%) | i1k Acc(%) | i21k Acc(%) |
|---|---|---|---|---|---|---|
| CNN | | | | | | |
| ResNetV2-50x1 | 76.86 | 79.2 | 97.63 | 97.87 | 86.04 | 88.97 |
| ResNetV2-101x1 | 77.99 | 81.28 | 96.9 | 98.8 | 83.03 | 91.12 |
| ResNetV2-50x3 | 79.13 | 82.77 | 98.02 | 98.91 | 86.69 | 91.53 |
| ResNetV2-101x3 | 79.59 | 83.56 | 97.51 | 99.09 | 84.93 | 92.65 |
| ViT | | | | | | |
| ViT-Ti/16 | 69.69 | 78.04 | 96.61 | 98.22 | 82.95 | 88.13 |
| ViT-S/16 | 78.38 | 83.64 | 98.15 | 98.96 | 87.52 | 92.15 |
| ViT-S/32 | 68.72 | 79.52 | 96.79 | 98.45 | 84.24 | 90.74 |
| ViT-B/16 | 78.1 | 86.06 | 99.28 | 99.29 | 94.02 | 94.02 |
| ViT-B/32 | 72.04 | 83.49 | 97.59 | 99.09 | 85.59 | 93.28 |
| ViT-L/16 | 74.6 | 85.46 | 98.79 | 99.5 | 90.5 | 94.92 |
| ViT-L/32 | - | 81.07 | - | 98.95 | - | 92.56 |
| Hybrid | | | | | | |
| ViT-R+Ti/16 | 66.98 | 75.12 | 95.73 | 97.37 | 78.39 | 85.48 |
| ViT-R26+S/32 | 78.18 | 83.77 | 97.14 | 98.8 | 82.85 | 91.82 |
| ViT-R50+B/16 | - | 84.92 | - | 99.03 | - | 91.07 |
| ViT-R50+L/32 | 73.52 | 86.13 | 97.37 | 99.23 | 79.69 | 92.85 |

## 4.2 Adversarial Robustness of CNNs, ViTs, and Hybrids

This section shows the adversarial robustness of CNNs, ViTs, and hybrids against white- and black-box attacks. In addition, the results are visualized according to the evaluation and pretraining datasets for a fair comparison.

### 4.2.1 White-Box Attack

Figure 1 shows the white-box attack results using the FGSM attack. For benign images, all three types of models performed similarly on all types of datasets. However, for adversarial examples, ViTs and hybrids generally had better classification performance than CNNs. Even if epsilon, the size of the perturbation, was changed, the adversarial robustness of CNNs was generally lower than that of other structures. It can be seen that ViTs and hybrids had larger performance differences according to their capacity than CNNs. CNNs showed vulnerable results when pretrained on ImageNet-1K but significantly improved robustness when pretrained on ImageNet-21K. Even when the epsilon was large, CNNs performed better than ViTs and hybrids. The PGD attack results in Figure 2 show similar results. However, none of the three types of models showed no a significant difference from the FGSM results.

We compared the loss of each model for the adversarial examples to derive more meaningful results because the defensive performance of the models converged to zero when the epsilon was 0.01 or greater. For this reason, model loss and accuracy were compared using epsilon values of 0.01, 0.03, and 0.1. Figure 3 shows the accuracy (dashed line) and the mean loss (solid line) for the adversarial examples of the models. As seen in the figure, at epsilon = 0.03, most of the models' accuracies converged to zero. However, CNNs had a larger loss than the other models. When ImageNet-1K was used as the pretraining dataset, hybrids had the lowest loss, followed by ViTs and CNNs. Through these results, we assumed that the lower the loss of a model on adversarial examples compared to other models, the harder it is to optimize adversarial examples to degrade the performance of the model. To verify this assumption, we compared the change in loss and accuracy with respect to the change of the attack steps by applying PGD with epsilon 0.01, where

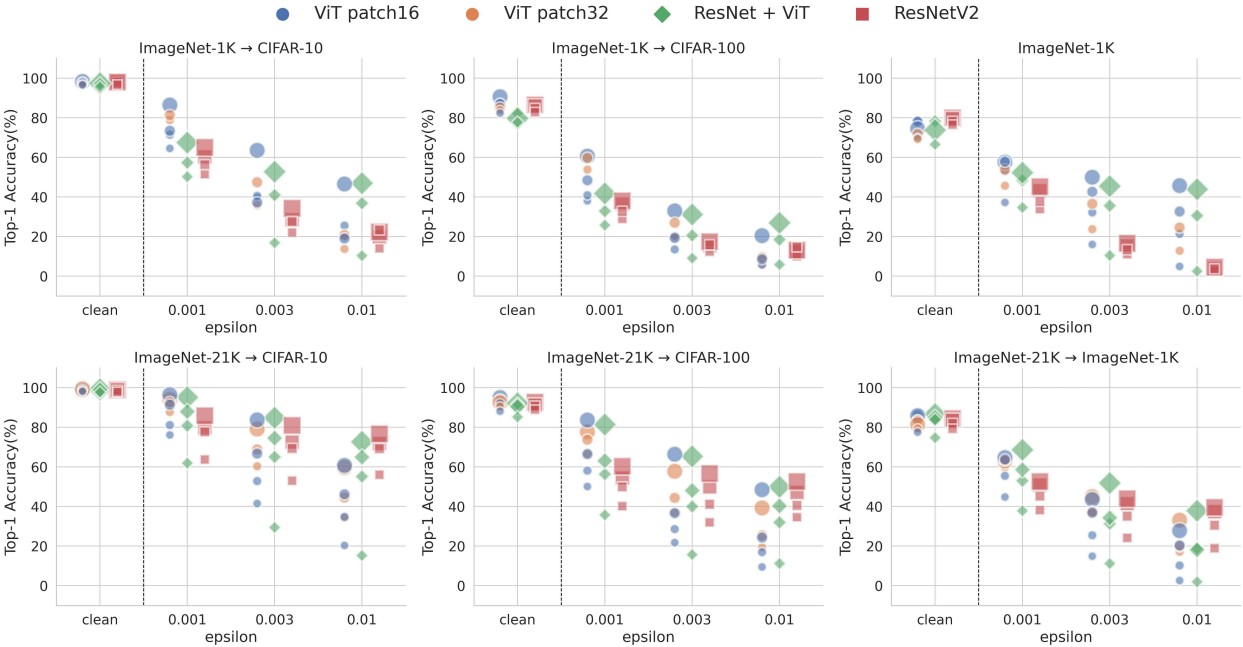

Figure 1: Top-1 accuracy (%) of all models on the adversarial examples generated by FGSM attacks. The x-axis in the graph represents the clean accuracy of a subset of the dataset and the accuracy by epsilon size. From the left, the columns correspond to CIFAR-10, CIFAR-100, and ImageNet-1K as the evaluation datasets. The rows correspond to ImageNet -1K and ImageNet-21K as the pretraining datasets.

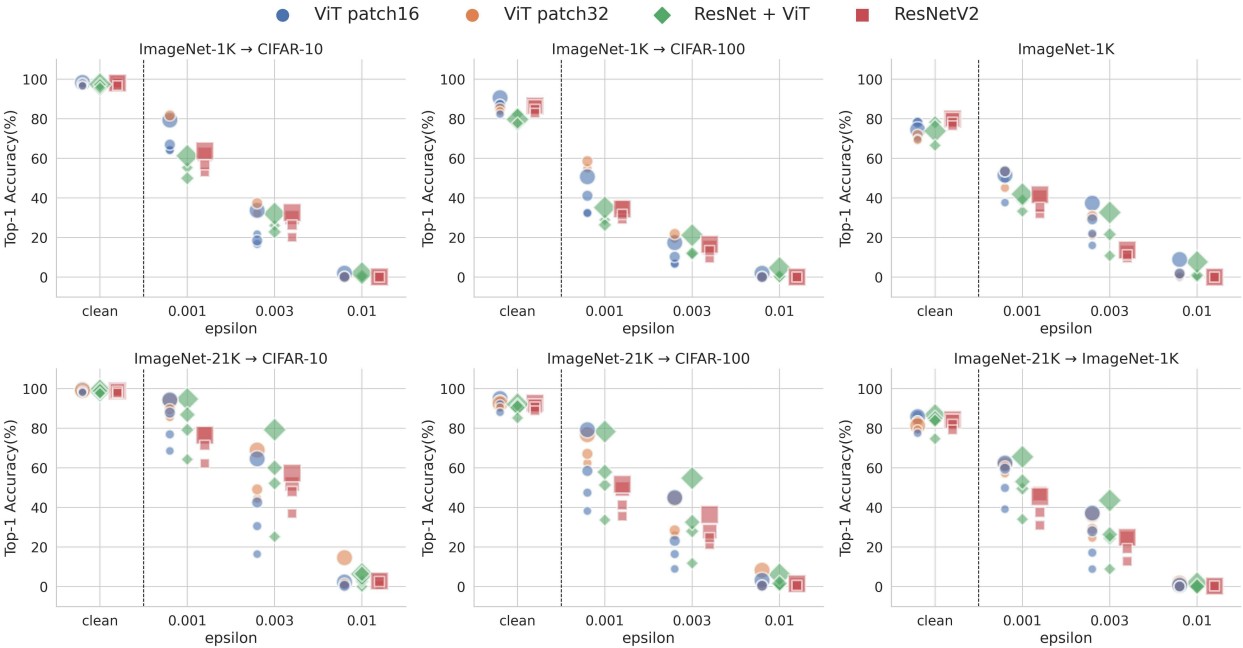

Figure 2: Top-1 accuracy (%) of all models on the adversarial examples generated by PGD attacks. The x-axis in the graph represents the clean accuracy of a subset of the dataset and the accuracy according to epsilon size. From the left, the columns correspond to CIFAR-10, CIFAR-100, and ImageNet-1K as the evaluation datasets. The rows correspond to ImageNet-1K and ImageNet-21K as the pretraining datasets.

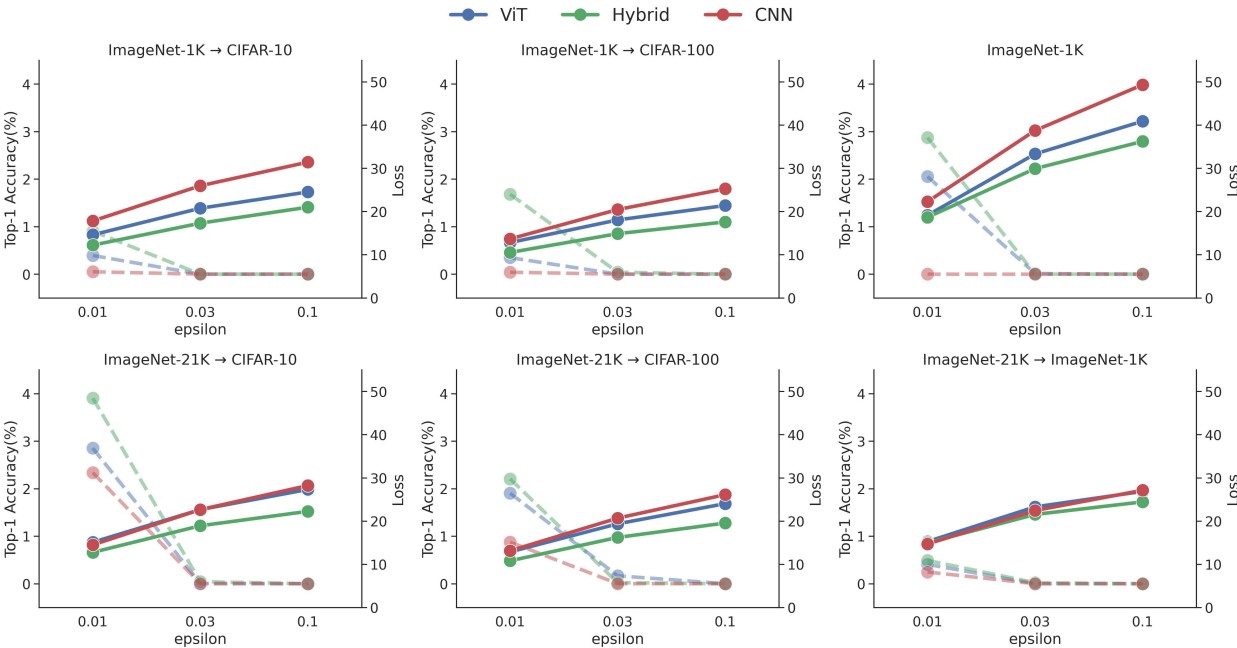

Figure 3: Comparison of the models' loss and top-1 accuracy(%) in adversarial examples at epsilon = 0.01, 0.03, and 0.1. The dotted lines in the graphs refer to the top-1 accuracy (%), and the solid lines refer to the loss. From the left, the columns correspond to CIFAR-10, CIFAR-100, and ImageNet-1K as the evaluation datasets. The rows correspond to ImageNet-1K and ImageNet-21K as the pretraining datasets.

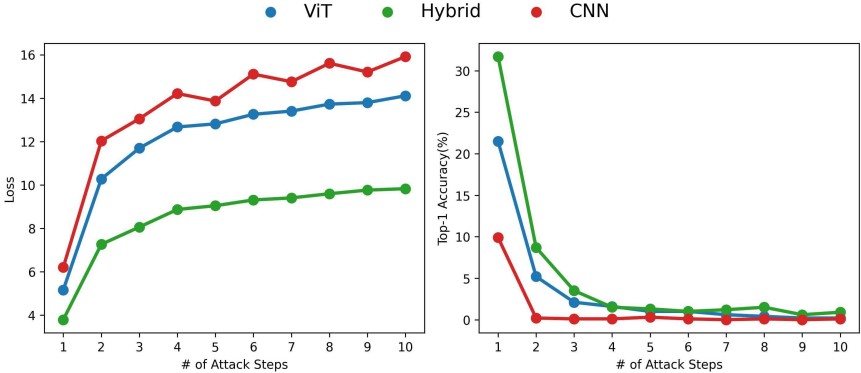

Figure 4: Comparison of the models' loss and accuracy in adversarial examples according to attack steps. ViT is ViT-S/32, Hybrid is ViT-R26+S/32, and CNN is ResNetV2-50x1. The evaluation data is CIFAR-10. PGD with epsilon 0.01 is used as adversarial attack

all models' accuracy converges to zero. We selected ViT-S/32, ViT-R26+S/32, and ResNetV2-50x1 models, each with a similar number of parameters, for comparison. The strong White-box attacks introduced earlier were optimized through 40 attack steps, making it difficult to compare the model's robustness as epsilon increases. However, by comparing the changes in loss and accuracy as a function of attack steps in Figure 4, we can see that the hybrid model with the smallest loss in Figure 3 also showed the smallest change in loss, requiring more optimization for adversarial examples to degrade the model's performance. ViT and CNN followed the Hybrid in terms of optimization costs for an adversarial attack. To summarize, the other two types require larger epsilon and more optimization steps with PGD compared to CNN.

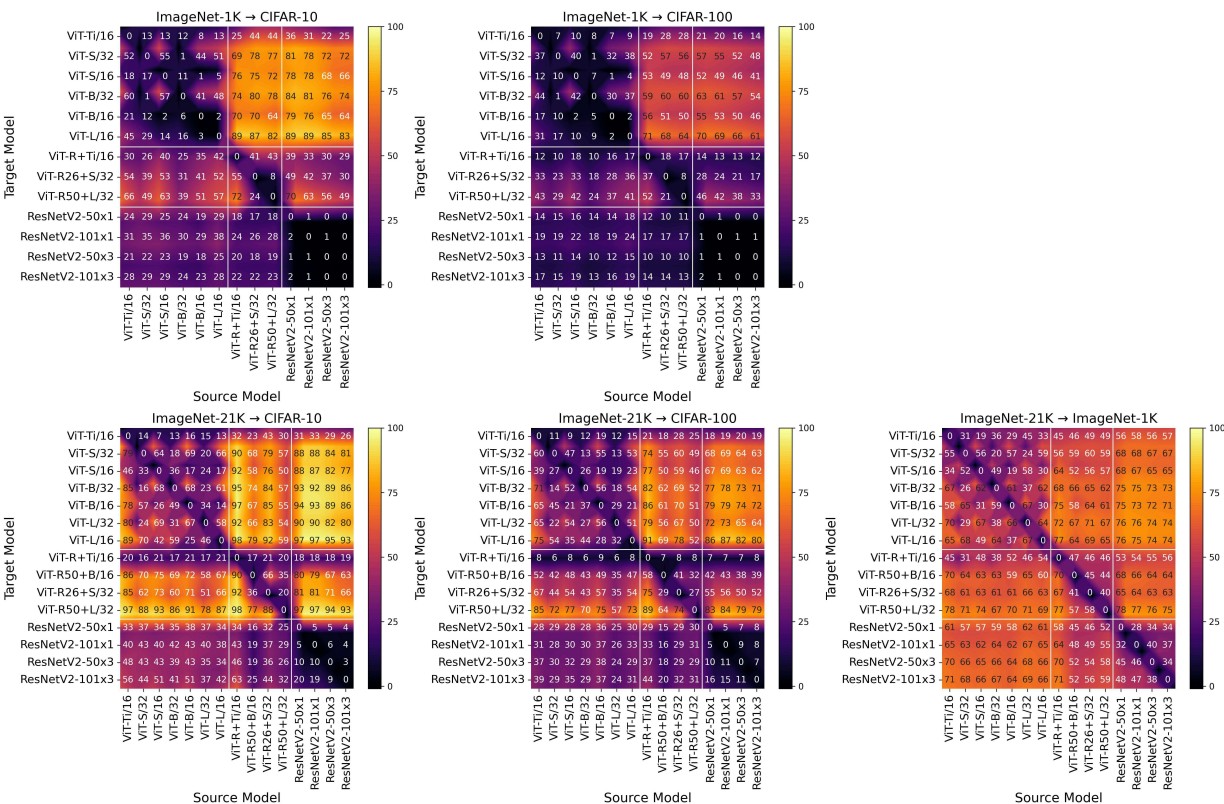

Figure 5: Top-1 accuracy (%) of the models on the adversarial examples generated between the models. The y-axis of the heatmap refers to the target models that evaluated the adversarial examples. The x-axis of the heatmap refers to the source models that generated the adversarial examples. From the left, the columns correspond to CIFAR-10, CIFAR-100, and ImageNet-1K as the evaluation datasets. The rows correspond to ImageNet-1K and ImageNet-21K as the pretraining datasets.

### 4.2.2 Black-Box Attack

To examine the transferability of adversarial examples between the models, we evaluated their defensive performance against adversarial examples generated from different models with epsilon = 0.1. Therefore, the models' losses and accuracies were compared for epsilon values of 0.01, 0.03, and 0.1. Figure 5 shows heatmaps that visualize the defensive performance of the models against the transfer attacks. The number in each cell represents the classification accuracy when the adversarial examples generated in the corresponding model on the x-axis were input in the corresponding model on the y-axis. The diagonal elements represent the defensive performance against the white-box attacks because the model that generated the adversarial examples and the model that evaluated it was the same.

When compared based on the evaluation datasets, the models evaluated on CIFAR-10 and CIFAR-100 showed a significant difference in robustness against black-box attacks of each model type. ViTs had low defensive performance against the adversarial examples generated by other ViTs, but their defensive performance was high against those generated from hybrids and CNNs. Furthermore, as the model capacity increased, their robustness also increased, and the same trend was confirmed when the input patch size was large. As with the experimental results for the white-box attacks, we found that ViTs exhibited higher robustness, even when the models were pretrained using ImageNet-21K. In contrast, the robustness of hybrids to adversarial examples generated in ViTs and CNNs improved more when ImageNet-21K was used as the pretraining data than when ImageNet-1K was used. However, the CNNs all had low defensive performance against not only

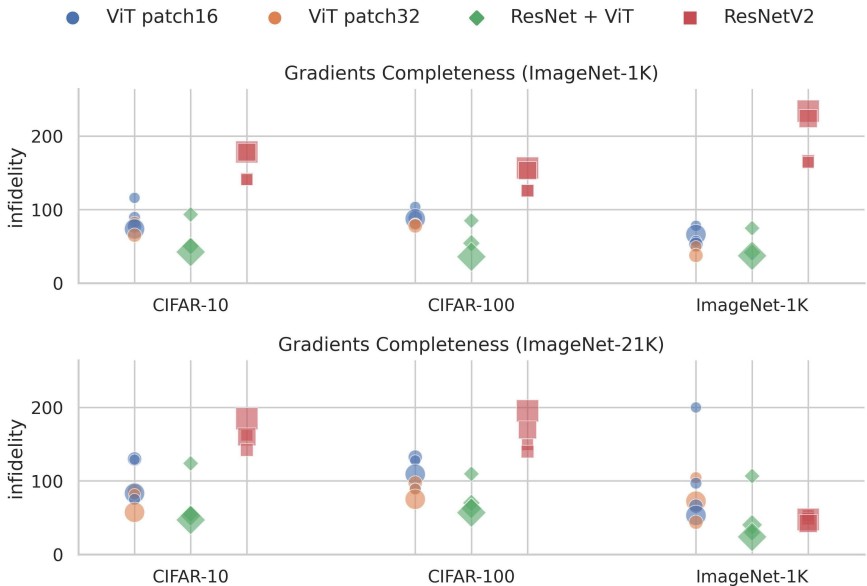

Figure 6: Infidelity caused by random noise using the vanilla gradient as the attribution method. The x-axis corresponds to CIFAR-10, CIFAR-100, and ImageNet-1K as the evaluation datasets. The rows correspond to ImageNet-1K and ImageNet-21K as the pretraining datasets.

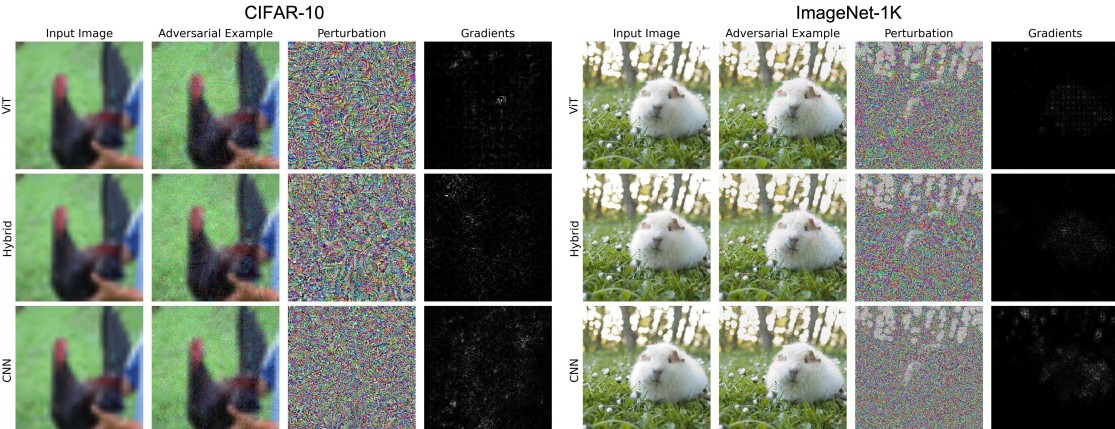

Figure 7: Visualization of the adversarial examples and the attributions generated by the vanilla gradient. The rows show the results of ViT, hybrid, and CNN results from top to bottom. The columns contain the input images, adversarial examples, perturbations, and saliency maps created by the vanilla gradient in order from left to right.

the adversarial examples generated from the same CNN-type models but also those generated from other models. To summarize, CNNs were more vulnerable to transfer attacks than ViTs.

# 5 The Quantitative and Qualitative Analysis

## 5.1 Analysis Using Infidelity

Figure 6 shows the infidelity of the models according to the pretraining and evaluation datasets. In this figure, CNNs showed greater infidelity than other models in almost all cases. On the other hand, ViTs had

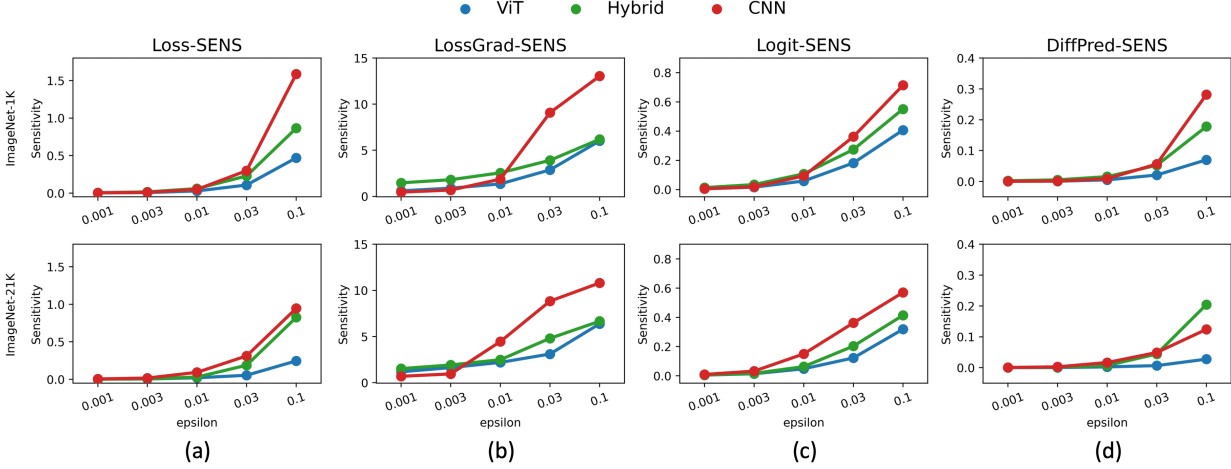

Figure 8: Comparison of proposed sensitivities on CIFAR-10. The rows correspond to ImageNet-1K and ImageNet-21K as the pretraining datasets. The columns correspond to Loss-SENS, LossGrad-SENS, Logit-SENS, and DiffPred-SENS as the evaluation metrics.

lower infidelity scores than CNNs, while hybrids had the lowest infidelity scores. From the model capacity perspective, CNNs had greater infidelity in complex models, while ViTs and hybrids showed lower infidelity. According to Yeh et al. (2019), these results suggest that ViTs have smaller and smoother gradients than CNNs. It can also be understood that hybrids, which are ViTs using features extracted by CNNs, enhance this characteristic.

Figure 7 visualizes adversarial examples, the perturbations, and the saliency maps for the images from CIFAR-10 and ImageNet-1K for a qualitative comparison between the three types of the model. The qualitative analysis is based on the differences in infidelity scores between the models shown in Figure 6. The adversarial examples are not distinguished from the input images. There is also no pattern to be found in the perturbations. However, the saliency maps generated by the vanilla gradients have a specific pattern. The ViT saliency maps focus on the objects in the input images. Moreover, the characteristics of the model that learned the inputs are reflected at the patch level. Similar to ViTs, the saliency maps of the hybrids also focus on the objects. However, they do not indicate the boundaries between patches. In the case of CNNs, the attributions in the saliency maps are distributed, and the boundaries between the objects and the backgrounds are unclear.

## 5.2 Sensitivity Difference Between CNNs, ViTs, and Hybrids

The proposed sensitivities to random noise using the CIFAR-10 dataset are shown in Figure 8. In Loss-SENS (a), which measures the sensitivity of the model's loss, the sensitivity of the CNNs became very large when the epsilon was greater than 0.03. In LossGrad-SENS (b), unlike the other metrics, the CNNs had smaller values than the other models with a small epsilon but recorded very high values for epsilon values of 0.01 and higher. In Logit-SENS (c), regardless of the pretraining dataset type, the trend was the same as with DiffPred-SENS when ImageNet-1K was used as the pretraining data. This means that the robustness of the CNNs dropped rapidly as the magnitude of the noise increased. In the case of DiffPred-SENS (d), which counts the cases showing different prediction results for normal and perturbed images, there was little difference between the three model groups for epsilon values up to 0.01. However, when epsilon was 0.1, the CNN DiffPred-SENS values were larger than the other models when pretrained using ImageNet-1K, but the CNNs had lower values than the hybrids when the models were pretrained using ImageNet-21K. Overall, when using ImageNet-21K, the model's sensitivity tended to be lowered, which was even more significant in the CNNs.

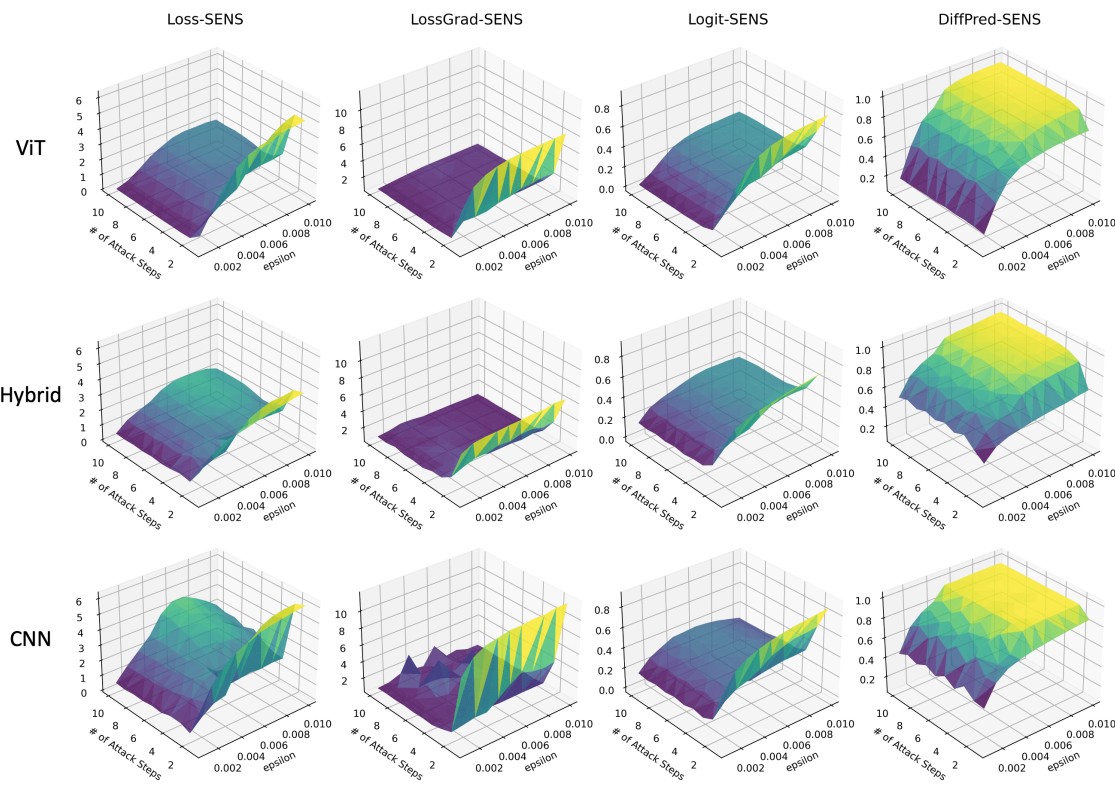

Figure 9: Proposed sensitivities to adversarial perturbation on the CIFAR-10 dataset. The rows represent the results of ViT, hybrid, and CNN results from top to bottom. The columns are the Loss-SENS, LossGrad-SENS, Logit-SENS, and DiffPred-SENS from left to right.

Figure 9 shows the measurements of the models' sensitivity to adversarial perturbation using the proposed metrics on CIFAR-10. For inference, the adversarial perturbation generated through the PGD attack was used. For a fair comparison, ViT-S/32, ViT+R26-S/32, and ResNetV2-50x1 were selected for the networks because of their similar model capacity. In DiffPred-SENS, it can be seen that different classes were predicted for normal data and adversarial examples by all model types when the epsilon and attack steps were large. However, the CNN-based model was vulnerable to adversarial perturbation, as the figure shows a yellow area for lower epsilons and attack steps. For the other metrics, the slope of the graph can be interpreted from an optimization point of view. If the slope is steep at a low attack step and stable afterward, the model is a vulnerable model that can be fooled with a small attack iteration. From this point of view, it can be seen that the CNN-based model, which shows a very steep slope at the first attack step in all metrics, is vulnerable to adversarial attacks. In particular, the CNN had a larger Loss-SENS value than the other model structures as the attack step was repeated, meaning that the CNN loss increased as the magnitude of the adversarial perturbation increased. After all, interpretation is difficult for strong attacks because the classification accuracy of all the models converged to zero, but in terms of loss, it can be seen that the CNN was more vulnerable than the other structures.

Figure 10 presents the loss landscape of the input space. We measured the model loss for the input space using two directions of sampled random noise from a normal distribution. The x- and y-axes in Figure 10 represent the values for scaling the two noise images. The two noise images scaled by the values of the x- and y-axes are added to the original image to become the final image for measuring the model's robustness. The z-axis is the loss value of the model when the corresponding image is input to the model. In other words, in the case of the center, the loss of the model is very low because it is the same as the original image, but the loss increases severely toward the edges as noise is added. Like Figure 9, ViT-S/32, ViT+R26-S/32, and ResNetV2-50x1 were selected for the networks for a fair comparison because the models had similar numbers

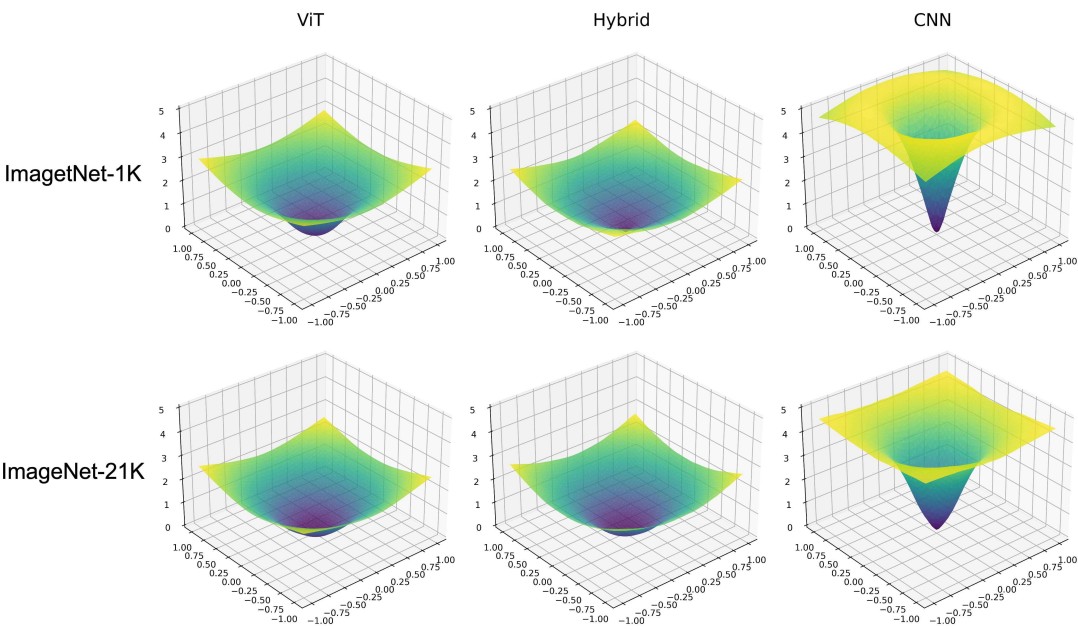

Figure 10: Visualization of the loss landscape in the input space. The rows correspond to ImageNet-1K and ImageNet-21K as pretraining datasets. The columns represent the results of ViT, Hybrid, and CNN from left to right.

of parameters. As can be seen in the figure, the CNN-based model performed poorly even with less noise than the ViT- and hybrid-based models. In other words, the CNN had a steep loss slope before and after a small change in the input. In addition, the increase in the loss was much larger than that of the other models. This result indicates that CNNs have an input space that is structurally very sensitive to noise. Even when the pretraining dataset was different, the same tendency was shown, although all models were more robust when pretraining with ImageNet-21K than with ImageNet-1K.

## 6 Conclusion

In this study, the adversarial robustness of CNNs, ViTs, and hybrids structures widely used in the computer-vision domain was measured and analyzed. For a fair comparison, structures of a similar size were used in each model category, and adversarial robustness was evaluated using both white- and black-box attacks. According to the findings, CNNs, with the exception of ImageNet-1K, are more vulnerable to adversarial attacks using CIFAR-10 and CIFAR-100. One interesting observation is that the difference between models was reduced in ImageNet-1K than CIFAR-10 and CIFAR-100. We suspect that this might be caused by the resolution of the original image, but a more systematic investigation needs to be conducted.

To further investigate why ViTs and hybrids indicate robustness compared to CNNs, we proposed four metrics to quantitatively measure the sensitivity of the model. We analyzed the models using infidelity and the proposed metrics. Through the proposed metrics, our experiments showed that the output of CNNs changes significantly compared to other models for the perturbed input. In addition, they have more sensitive the loss landscape in the input space. As a result, we found that CNNs are more sensitive to random noise and adversarial perturbation than other structures. It is hoped that the experiments conducted in this study will be the basis for further understanding and trusting the models.

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
