# OpenReview forum: "Why Do Vision Transformers Have Better Adversarial Robustness than CNNs?"
_TMLR — Rejected by TMLR_

### Review · Reviewer_csDh · 2023-01-06

**Summary Of Contributions:**

This work analyses the robustness (noise, adversarial) of common vision architectures (CNN, ViT, hybrid). Based on fidelity score, novel metrics are introduced to evaluate model robustness. Empirical findings are presented for the CIFAR10/100 and ImageNet-1K and ImageNet-21K datasets addressing robustness (white-box attacks), transferability (black-box), and evaluation using the novel metrics.

**Audience:**

Yes

**Broader Impact Concerns:**

There are no ethical implivations of this work.

**Claims And Evidence:**

No

**Requested Changes:**

Addressing the two major concerns is critical for acceptance, since they address central points / claims of the work.



**Strengths And Weaknesses:**

Strengths

The work addresses the important topic of analysing robustness for common vision architectures. It provides four novel metrics as well as some empirical evaluation.


Weaknesses

A first major concern is about the proposed metrics (LossGrad-SENS, Logit-SENS, DiffPred-SENS, Loss-SENS): The metrics appear redundant and no specific cases to highlight their individual utability are provied. Why are these four metrics necessary? In which cases are these specifically informative?

My second major concern is about the empirical results and especially the chosen datasets. The main finding of the paper is that ViT are somewhat more robust compared to CNN models. Considering the results (esp. Fig. 1, 2, 4, 5) this seems primarily true for evaluations on the CIFAR10/100 datasets. Yet, this effect seems to decrease for the ImageNet21 dataset. An under evaluated factor here seems to be the size and maybe resolution of the used dataset, which needs to be further investigated to harden the claims or give proper constraints.

---

> ### Author Response · Authors · 2023-01-31
> **Authors reply to reviewer csDh**
>
> Thank you for your valuable comments. The detailed response and the revised part in the paper for each comment is as follows.
>
> **Revised Contents:**
>
> **Weakness1. A first major concern is about the proposed metrics (LossGrad-SENS, Logit-SENS, DiffPred-SENS, Loss-SENS): The metrics appear redundant and no specific cases to highlight their individual usability are provided. Why are these four metrics necessary? In which cases are these specifically informative?**
>
> We added an explanation about why the difference in the model's robustness against adversarial attack occurs and why we compared not only infidelity but also four evaluation metrics in Section 3.2.2 Proposed Metrics (pages 5-6). The added content includes the reason for using infidelity for an objective evaluation of adversarial robustness and what is lacking when using only infidelity, and the reasons for using the four evaluation metrics. The reasons for using the four evaluation metrics can be divided into perspectives of loss and target, and we described what results can be confirmed by comparing the four together.
>
> **Weakness2. My second major concern is about the empirical results and especially the chosen datasets. The main finding of the paper is that ViT are somewhat more robust compared to CNN models. Considering the results (esp. Fig. 1, 2, 4, 5) this seems primarily true for evaluations on the CIFAR10/100 datasets. Yet, this effect seems to decrease for the ImageNet21 dataset. An under evaluated factor here seems to be the size and maybe resolution of the used dataset, which needs to be further investigated to harden the claims or give proper constraints.**
>
> We also agree that there is a question about the difference in robustness between models being clear only in CIFAR10/100 and becoming less noticeable in ImageNet1K. Although CIFAR10/100 originally has 32x32 resolution, to ensure fair comparison, we used pre-trained models and structured it to have the same resolution as ImageNet1K, which is 224x224 by performing bilinear interpolation on the 32x32 resolution (4.1 Experimental Settings - second paragraph, page 6). Hence, we think that the decrease in model robustness difference in ImageNet1K is due to increased image resolution. However, we were unable to confirm a clear reason, so we mentioned it as a limitation of current study in section 6, Conclusion.

---

### Review · Reviewer_aUxf · 2023-01-09

**Summary Of Contributions:**

This paper tries to compare the adversarial robustness of ViTs and CNNs in a more fair manner and study the reason behind the difference. The paper uses several model-sensitivity metrics to compare the robustness and shows that ViTs are more robust than CNNs. However, the use of these metrics are not well justified and the experiments are not quite sound.

**Audience:**

Yes

**Claims And Evidence:**

No

**Requested Changes:**

- Justify the use of the sensitivity metrics for adversarial robustness.
- Tailor the attack methods for the sensitivity metrics.
- More clearly explain the difference with existing works and contributions of this paper.


**Strengths And Weaknesses:**

Strengths are unclear for now, as the differences with existing works  (the use sensitivity metrics and different perturbation sizes) are not reasonably justified yet and their soundness are questionable.

Weaknesses:
- This paper studied adversarial robustness, but it is confusing that the paper suddenly introduced metrics for interpretation rather than adversarial robustness by citing Yeh et al., 2019. It is unclear why this paper uses those sensitivity metrics to compare the robustness, which is not a typical way to compare the adversarial robustness, yet it is not justified in this paper.
- Even if the authors could justify the use of the sensitivity metrics, the attack methods need to be modified to consider the worst-case sensitivity. Regular PGD attacks aim to find the worst-case adversarial accuracy, which is not optimized towards finding the perturbation that leads to the worst-case sensitivity.
- This paper criticized that Bhojanapalli et al. (2021) “failed to show the difference between the two models depending on the size of the adversarial perturbations”. First, I think it is not suitable to say “failed to” unless Bhojanapalli et al. (2021) has tried that. Also, I doubt if experiments in this paper using larger perturbation radii is reasonable, because the adversarial accuracies of all the models are all extremely low (<=4% in Figure 3) and the comparison does not seem to be meaningful.

---

> ### Author Response · Authors · 2023-01-31
> **Authors reply to reviewer aUxf**
>
> Thank you for your valuable comments. We think that it may be difficult to understand our proposed method and experiment for evaluating adversarial robustness due to a lack of explanation. The detailed response and the revised part in the paper for each comment is as follows.
>
> **Revised contents:**
>
> **Weakness 1. This paper studied adversarial robustness, but it is confusing that the paper suddenly introduced metrics for interpretation rather than adversarial robustness by citing Yeh et al., 2019. It is unclear why this paper uses those sensitivity metrics to compare the robustness, which is not a typical way to compare the adversarial robustness, yet it is not justified in this paper.**
>
> We added an explanation about why the difference in the model's robustness against adversarial attack occurs and why we compared not only infidelity but also four evaluation metrics in Section 3.2.2 Proposed Metrics (pages 5-6). The added content includes the reason for using infidelity for an objective evaluation of adversarial robustness and what is lacking when using only infidelity, and the reasons for using the four evaluation metrics. The reasons for using the four evaluation metrics can be divided into perspectives of loss and target, and we described what results can be confirmed by comparing the four together.
>
> **Weakness 2. Even if the authors could justify the use of the sensitivity metrics, the attack methods need to be modified to consider the worst-case sensitivity. Regular PGD attacks aim to find the worst-case adversarial accuracy, which is not optimized towards finding the perturbation that leads to the worst-case sensitivity.**
>
> We agree that the general adversarial attack you mentioned is aimed at reducing adversarial accuracy. However, our proposed method aims to compare the sensitivity of the model to adversarial examples generated through adversarial attacks commonly applied to the model. Our goal is to understand the robustness of the model's performance, not to optimize the evaluation metric dependent on adversarial attacks, as that would be in a different direction from our goal of understanding the robustness of the model's sensitivity.

---

> ### Author Response · Authors · 2023-01-31
> **Authors reply to reviewer aUxf**
>
> **Weakness 3. This paper criticized that Bhojanapalli et al. (2021) “failed to show the difference between the two models depending on the size of the adversarial perturbations”. First, I think it is not suitable to say “failed to” unless Bhojanapalli et al. (2021) has tried that. Also, I doubt if experiments in this paper using larger perturbation radii is reasonable, because the adversarial accuracies of all the models are all extremely low (<=4% in Figure 3) and the comparison does not seem to be meaningful.**
>
> First, upon revisiting the prior study by Bhojanapalli et al. (2021) for comparison with our research, it was confirmed that the paper did not compare the results based on perturbation size during adversarial attack experiments. Bhojanapalli et al. (2021) mainly performed adversarial robustness for ViT and CNN based on the model size and pre-trained data, and confirmed how information flows within the model through an ablation study of the ViT structure. Perturbation size is one of the factors that significantly affects the performance of the model in adversarial attacks. Therefore, it is also important to compare how models change based on perturbation size when evaluating adversarial robustness. Hence we revised the expression “failed to show the difference between the two models depending on the size of the adversarial perturbations” to “However, they did not provide sufficient demonstrations regarding the disparities between the two models with respect to the magnitudes of adversarial perturbations.”
>
> Next, in Figure 3, we compared the loss and accuracy between models when large epsilons wre used contraty to Figures 1 and 2. However, when epsilon increases beyond 0.03, accuracy may converge to 0, making it appear that comparing adversarial robustness is meaningless. Nevertheless, through the results of Figure 3, we were able to see the difference in adversarial robustness from two perspectives, epsilon size and attack step. The fact that a particular model has a smaller loss compared to other models means that it is harder to optimize for adversarial examples, thus deteriorating its performance.
> First, as we can see from Figures 2 and 3, ViT and Hybrid have relatively smaller performance degeneration compared to CNN in terms of adversarial attack as the size of epsilon increases. Therefore, even with small epsilon, CNN has a relatively larger performance degeneration compared to the other two models, while the other ViT models need to use a larger epsilon to worsen the performance.
> Next, we added Figure 4 to compare from the perspective of the attack step. Figure 4 represents the change in model loss and accuracy by repeating the PGD attack 10 steps with epsilon 0.01 from 1. As can be seen from the results, CNN quickly changes when optimizing for adversarial examples compared to the other two models, while the other two models converge more slowly. Thus, we can confirm that the ViT models require more cost for adversarial attack than CNN.

---

### Review · Reviewer_wdA8 · 2023-01-17

**Summary Of Contributions:**

1. This paper evaluates the adversarial robustness of Convolutional Neural Networks, Vision Transformers and hybrid models of CNNs and ViTs.
2. The authors first evaluate the adversarial robustness of multiple models using existing methods like FGSM and PGD.
3. The authors then propose four different metrics based on max-sensitivity to measure adversarial robustness.

**Audience:**

Yes

**Broader Impact Concerns:**

I have no broader impact concerns.

**Claims And Evidence:**

Yes

**Requested Changes:**

1. More explanation and justification for the proposed metrics, differences among them.
2. Extend this analysis to more CNN/ViT architectures to generalize the findings.

**Strengths And Weaknesses:**

Strengths:
1. The paper is well-written and easy to follow.
2. The adversarial analysis is comprehensive.

Weaknesses:
1. My major concern is the lack of proper explanation on the proposed four different metrics. The authors should justify how those metrics can help measure adversarial robustness. Right now, there is just a statement in the paper "Additionally, based on the max-sensitivity (Yeh et al., 2019),
we propose DiffPred-SENS, Loss-SENS, LossGrad-SENS, and Logit-SENS".
2. It is unclear how Logit-SENS differ from Loss-SENS as the models' loss is calculated using logits.
3. Another major concern is the universality of the findings to CNNs and ViTs. Since, the analysis is performed with just a single CNN/ViT architecture, how can we assume these findings will be applicable to all CNN/ViT architectures.

---

> ### Author Response · Authors · 2023-01-31
> **Authors reply to reviewer wdA8**
>
> Thank you for your valuable comments. We agree that the paper lacked specific explanations about the proposed content and experiment. Based on your comments, we added additional descriptions about the motivations and reasons for the proposed method for an objective evaluation of adversarial robustness. The responses to the contents mentioned as weakness are as follows.
>
> **Revised Contents:**
>
> **Weakness 1. My major concern is the lack of proper explanation on the proposed four different metrics. The authors should justify how those metrics can help measure adversarial robustness. Right now, there is just a statement in the paper "Additionally, based on the max-sensitivity (Yeh et al., 2019), we propose DiffPred-SENS, Loss-SENS, LossGrad-SENS, and Logit-SENS".**
>
> We added an explanation about why the difference in the model's robustness against adversarial attack occurs and why we compared not only infidelity but also four evaluation metrics in Section 3.2.2 Proposed Metrics (pages 5-6). The added content includes the reason for using infidelity for an objective evaluation of adversarial robustness and what is lacking when using only infidelity, and the reasons for using the four evaluation metrics. The reasons for using the four evaluation metrics can be divided into perspectives of loss and target, and we described what results can be confirmed by comparing the four together.
>
> **Weakness 2. It is unclear how Logit-SENS differ from Loss-SENS as the models' loss is calculated using logits.**
>
> The two indices are proposed to evaluate sensitivity from different perspectives. Loss-SENS is an index to check how sensitively the model's loss changes, and Logit-SENS is an index to check how the size of logits changes. Accordingly, as can be seen in Figures 8 and 9, the trends shown by the two indices can be seen to be different. In Figure 8 (a), we can see that Loss-SENS dramatically rises when the epsilon is 0.1. In Figure 8 (c), on the other hand, we can see that Logit-SENS increases continuously as the epsilon increases, contrary to Loss-SENS. Logit-SENS significantly changes in size when the confidence of the target class largely changes.
>
> Hence, from the perspective of the model's sensitivity, the change in target confidence continually changes as epsilon increases, but the loss changes dramatically. When comparing between models, the ViT model's change in logit is continuously increasing, similarly to the other two models, but the loss has relatively little change when epsilon is 0.1.
>
>
> **Weakness 3. Another major concern is the universality of the findings to CNNs and ViTs. Since, the analysis is performed with just a single CNN/ViT architecture, how can we assume these findings will be applicable to all CNN/ViT architectures.**
>
> We understand your concern in some perspective. However, it has been difficult to systematically control experiments comparing all CNN and ViT models. Our planned experimental conditions involve using models of similar sizes and trained with the same pre-training data for CNN, hybrid, and ViT models. Therefore, we conducted comparative experiments using representative models used in each type. Although our experimental system is not the same, the results of adversarial attack on CNN and ViT can be seen in other previous studies [1, 2]. Based on the similarity of the results obtained from the present case with the results of adversarial attack we conducted, it can be assumed that other CNN and ViT models are also similar to the results of the Quantitative and Qualitative Analysis in Section 5.
>
> [1] Understanding robustness of transformers for image classification. ICCV 2021
> [2] On the adversarial robustness of vision transformers. TMLR 2022

---

### Decision · Action_Editors · 2023-02-17

**Recommendation:** Reject

**Comment:**

This paper evaluates the adversarial robustness of Convolutional Neural Networks, Vision Transformers and hybrid models of CNNs and ViTs. The authors first evaluate the adversarial robustness of multiple models using existing methods like FGSM and PGD. The authors then propose four different metrics based on max-sensitivity to measure adversarial robustness.

This paper has its own merits to this area. For example, the adversarial analysis is comprehensive. Moreover, we appreciate the authors for taking time for providing clarifications and rebuttal. However, three anonymous reviewers unanimously point out some critical yet unaddressed concerns after reading the rebuttal. For example, they all believe that the justification for the proposed metrics provided by the authors is not sufficient. The current paper says "it is necessary to determine whether the model has small and smoothed gradients to perturbation to evaluate the robustness of the model", which is a quick assertion without any justification. Also, they are not convinced why adversarial robustness should be evaluated via gradients instead of just the predictions of the model. Moreover, they have some major concern about the empirical results and especially the chosen datasets.

Hence, based on three negative recommendations, it is quite pity that I should recommend reject. However, we do believe that the updated version should be strong, and hope that the authors can update a new version to try other good venues.

**Audience:**

Yes

**Claims And Evidence:**

Yes